# Nanoparticle Detection on SEM Images Using a Neural Network and Semi-Synthetic Training Data

**DOI:** 10.3390/nano12111818

**Published:** 2022-05-26

**Authors:** Jorge David López Gutiérrez, Itzel Maria Abundez Barrera, Nayely Torres Gómez

**Affiliations:** División de Estudios de Posgrado e Investigación, Instituto Tecnológico de Toluca, Tecnológico Nacional de México, Av. Tecnológico s/n, Colonia Agrícola Bellavista, Metepec, México City 52149, Mexico; m20280262@toluca.tecnm.mx (J.D.L.G.); iabundezb@toluca.tecnm.mx (I.M.A.B.)

**Keywords:** scanning electron miscroscopy, nanoparticle detection, neural networks, synthetic data, yolov3, yolov4

## Abstract

Processing images represents a necessary step in the process of analysing the information gathered about nanoparticles after characteristic material samples have been scanned with electron microscopy, which often requires the use of image processing techniques or general purpose image manipulation software to carry out tasks such as nanoparticle detection and measurement. In recent years, the use of networks has been successfully implemented to detect and classify electron microscopy images as well as the objects within them. In this work, we present four detection models using two versions of the YOLO neural network architectures trained to detect cubical and quasi-spherical particles in SEM images; the training datasets are a mixture of real images and synthetic ones generated by a semi-arbitrary method. The resulting models were capable of detecting nanoparticles in images different than the ones used for training and identifying them in some cases as the close proximity between nanoparticles proved a challenge for the neural networks in most situations.

## 1. Introduction

Processing scanned electron microscopy (SEM) images is a commonly performed to study the results of nanomaterials synthesis processes in which the measuring and identification of objects provide the materials scientist with valuable morphological data about the material of interest. Typically processing SEM imagery was performed using image processing techniques tailored around the characteristic features of a given material such as shape, size, brightness and contrast difference between the observed objects [1,2]. Another approach was to use general purpose image processing applications such as imageJ to perform the analysis of electron microscopy images in a semi-automated manner, again making use of the knowledge of the defining characteristics of the specific material being studied [3]. While these methods yield precise results, they are not scalable.

In recent years, there have been works using convolutional neural networks (CNN) which have successfully classified electron microscopy images. Similarly, CNNs can be used for the classification of nanoparticles (NPs) in SEM images as demonstrated by the annotated image repository produced by members of Nanoscience Foundries and Fine Analysis Europe (NFFA), which was created by using retrained models of Google’s Inception family of CNNs [4,5]. Neural networks have been used previously in conjunction with traditional image processing algorithms to detect nanoparticles in images created through different electron microscopy techniques, not just SEM, as is the case for ParticlesNN [6], a web service designed to perform the detection and instance segmentation of platinum and palladium nanoparticles on images obtained through scanning tunneling microscopy (STM) as well as SEM or transmission electron microscopy (TEM), to name a few, by making use of Cascade Mask-RCNN neural networks [7]. Similarly, Kim et al. implemented the previously trained Imagenet database [8], Inception-v3 neural network, to detect core-shell type nanoparticles on SEM micrographs with a computer program named LIST [9]. Particularly, the YOLO convolutional neural network has been implemented previously with the purpose of detecting different structures in TEM images, not only particles such as cavities or other deformations on the surface of a specimen that has been radiated with an ion beam [10]. Because YOLO excels at real time detection, it has been used for in situ detection and tracking of objects found in a video feed from a transmission electron microscope [11]. It can also be used in tandem with other CNN architectures to obtain an improved instance segmentation model in one-dimensional nanostructures such as ZnO nanowire arrays [12].

The training of neural networks requires large amounts of data to be carefully annotated before the learning to yield better results. The acquisition and annotation of data can be partially automated when a method to artificially increase the available data is used. This synthetic data allows for the controlled balancing of classes in the dataset and frees the short amounts of real data to be used to evaluate the learning process of a neural network; a common approach to the creation of synthetic data is the rendering of 2D and 3D nanoparticle shapes and bodies, respectively, that resemble their real life counterparts [13,14].

The goal of this work was to implement a detection model based on the YOLO neural network aimed at detecting cubic and quasi-spherical nanoparticles in SEM images that are scattered across the scene. To be able to procure the annotated data necessary for the training of the neural network, a computer program was developed with the main function of artificially generating digital images that resemble SEM micrographs depicting Ag nanoparticles from the original sample.

## 2. Materials and Methods

In this work, Ag nanocubes were previously synthesized at the UTSA using the polyol method in the following manner: heat 5ml of ethylene glycol (EG) up to 160 C. A 0.1 M silver nitrate ($AgNO_{3}$) dissolution was prepared in the EG and another 0.15M PVP one. Once the EG reached the desired temperature, the $AgNO_{3}$ and polyvinylpyrrolidone (PVP) dissolutions were added drop by drop, maintaining a magnetic stirring for 40 s. After this, it was allowed to cool down, and the resulting product was washed with deionized water and acetone prior to use and characterization. Synthetic hidroxiapatite (HA) nanopowder was acquired from Sigma-Aldrich for the purpose of obtaining SEM images depicting quasi-spherical particles.

### 2.1. Image Acquisition

Ag nanocubes and HA nanopowders were dispersed in isopropyl alcohol, and a drop was placed on a silicon flake for morphological characterization by scanning electron microscopy (SEM). This characterization was carried out at the Laboratory of Environmental Engineering Research of the Instituto Tecnológico de Toluca (ITT), the capture of SEM micrographs being performed using a JEOL JSM-6610LV operated at 20 kV with a resolution of 137 eV, a tungsten filament as the electron source and equipped with EDS.

### 2.2. YOLO Neural Network

Nanoparticle detection was performed through the use of YOLO (You Only Look Once) neural networks Versions 3 and 4 in their faster (tiny) architectures typically used for applications intended for real-time object detection, although for this project its use is meant to detect scattered nanoparticles. The training of these neural networks and their evaluation were performed on a computer equipped with a NVIDIA Quadro RTX 3000 with 6 GB VRAM, property of the ITT and using the YOLO implementations made with darknet [15,16].

### 2.3. Synthetic Images

The available micrographs of Ag nanocubes lacked sufficient numbers of dispersed particles to successfully train the neural network, causing a need for more images or a method to artificially increase the data. A method was proposed to increase the amount of available data for training. Since Ag nanocubes are not magnetic in nature and scatter arbitrarily across the sample holder without following any particular pattern, a simple computer program was developed to artificially generate micrographs containing scattered Ag nanocubes. This program’s logic tries to approximate said behaviour by placing them semi-arbitrarily in a new image that can be labelled and used for training.

The proposed method makes use of a set of 8 SEM micrographs with scattered nanoparticles. From each of these images, a series of NPs were selected and manually segmented. That is, a smaller image was obtained from a region of the original one; we refer to these segmented images as *templates*, and a group of templates extracted from the same image is called a *pattern*. The selected nanoparticle templates extracted from the source micrographs are of different sizes (in pixels) and orientation, thus creating diverse patterns for generating images. Figure 1 shows four templates selected from the micrograph displayed in Figure 2, the template images have a resolution of 71 × 68 pixels compared to 1280 × 960 px of the original source image; for this particular pattern Templates #2 and #3 (Figure 1b,c) are the same image as we wanted to test if repeating the template would have a negative effect during the training process. We did not observe such thing, and it was left that way as part of the pattern for synthetic image generation. Each pattern has an associated *pattern file*, a plain text file that contains the resolution of the original SEM micrograph and the filename of each template image; the information contained in this file will be used by the computer program to generate the synthetic images. Similarly, templates for quasi-spherical nanoparticles such as the ones shown in Figure 3 were obtained with the purpose of providing the training dataset with examples of quasi-spherical particles different from the ones present in the repository by the NFFA; Figure 4 shows one of the images from which templates of this morphology were selected.

Table 1 shows a description of the 8 different patterns used to create artificial micrographs by the previous algorithm, each one containing a different number of template images, as well as the difference in the type of morphologies present in them, with four patterns limited to cubic particles. All the original micrographs were captured at UTSA using a FEG Hitachi S-5500 ultra-high-resolution electron microscope (0.4 nm at 15 kV) with a BF/DF Duo STEM detector; and they all depict the same Ag nanocube sample which possesses a similar average length. In this case, it ranges from 0.13 μm to 0.15 μm. A series of synthetic images generated using the method proposed and the patterns described (one per pattern) are shown in Figure 5; as stated in Table 1, only images generated using patterns 4 through 7 can include quasi-spherical particles (Figure 5d, Figure 5e, Figure 5f, Figure 5g, respectively). The notorious difference in size between the particles in Figure 5a and the rest of the examples is due to the original micrograph having been captured using a greater magnification.

With the patterns created, a computer program was written in C++ and using OpenCV, a computer vision and image processing library [17], simply called *nanocube generator*. This CLI utility requires 2 inputs to be provided to it in order to produce a synthetic image, the pattern file and the maximum number of NPs to include in the image. With these two parameters, the nanocube generator initiates its execution where an image of the same resolution as the original micrograph in the pattern is generated in-memory and filled with a dark background. We refer to this image as a background image or canvas. The next steps are to load the templates listed in the pattern file and semi-randomly select one of them, rotate it and place it on the canvas in a semi-random position. Then the program proceeds to select another template and repeat the last two steps, making sure the new template image does not overlap on the area occupied by an already placed NP template. The procedure is repeated until no space is left on the canvas to place more template images, or the maximum number of NPs indicated when calling the program is reached. All semi-random values used for the described operations are performed using a seed generated by the computer clock during execution time of the program. An example of an image generated using the pattern based on the image from Figure 2 is shown in Figure 5b.

### 2.4. Training Datasets

As previously mentioned, the proposed methodology classifies nanoparticles based on their morphology in two categories: cubes and quasi-spheres, these being chosen for being simple shapes. For the first class, the images used to create the training datasets are the ones generated by the program previously described. For quasi-spheres, most images were selected from the particles subset of the public repository by Aversa et al. [5] made of 3412 JPEG images with a 1024×768 resolution, choosing the ones with quasi-spherical particles having a diameter smaller than 200 nm. A small number of Ag quasi-spheres are also included in images generated by the developed program. Preprocessing of the images before labelling consisted of removing the information bar embedded in them; the resolution for the images was not adjusted prior to the introduction to the training session.

For the purposes of training, two image sets containing nanocubes and quasi-spherical particles were prepared. Table 2 shows the proportion of each particle class on the datasets. Set A is an unbalanced one as it is composed of only 450 cube objects against over twice the number of quasi-spheres; in contrast, Set B has an equal amount of objects of both classes, 1025 each.

Both sets of micrographs were prepared for training the YOLO detector by labelling, using the program labelImg [18], the particles in them in the proposed classes, omitting those which are partially occluded by other objects or the image border. Imageset A, comprised of 105 images, containing 450 labelled cube class objects and 1000 quasi-sphere ones was divided into training, testing and validation batches at a ratio of 75%, 15% and 15%, respectively; in comparison, set B, made up of 1025 objects of each class across 112 images,was divided into training and testing batches of 80% and 20%, respectively. Four models were trained, two based on YOLOv3-tiny and the remaining pair on YOLOv4-tiny. Table 3 is a summary of the architecture and the dataset used to train it; each architecture was trained on both datasets with 200,080 training batches and a learning rate of 0.001. The images were rescaled at the input layer to the default resolution of the YOLO architecture, 416 × 416 pixels. Figure 6a,b show one of the images used for training in its original resolution of 1280 × 960 and the scaled down version to the mentioned resolution, respectively. This resolution preserved the shape of the nanoparticles and since the YOLO neural network changes the image dimension at random during training, the detection model gets optimized to make predictions for different image sizes [19]; additionally, it decreased the video memory requirements during the training process.

### 2.5. Validation Datasets

To evaluate the performance of the trained detector models, three validation datasets were prepared, each one composed of 20 individual images, Table 4 shows the count of labelled class objects in each validation dataset. The micrographs comprising Sets I and II belong to the ones obtained at UTSA and from the repository made available by NFFA [5] for nanocubes and quasi-spheres, respectively; Dataset III, on the other hand, is made up of images captured at ITT and include Ag nanocubes (Figure 7i,j) as well as HA quasi-spheres (Figure 7k,l). Figure 7 shows some of the images that belong to these datasets. Figure 7a–d are part of Dataset I and include quasi-spherical nanoparticles of an undetermined material as well as scattered Ag nanocubes; similarly, Dataset II also contains these types of nanoparticles with the difference lying in the fact that the nanocubes depicted in these images are very close to each other, with many lying on top of one another as shown in Figure 7e,f. Figure 7c,d,g,h display only quasi-spherical nanoparticles, of an unknown series of materials and are also the micrographs with the highest magnification used to train the neural network.

After running a detection process using the trained models on each validation dataset, the Jaccard coefficient (also known as intersection over union or IoU) was used as a measure of the degree to which the detected area and the labelled one overlap each other, with a threshold of 0.5 used to determine whether a detection was positive.
(1)$$J\left(A,B\right)=\frac{\left|A\cap B\right|}{\left|A\cup B\right|}$$

Furthermore, the median average precision (mAP), using the script developed by Cartucho et al. [20], was also determined to evaluate the classification of detected objects for each one of the trained models.
(2)$$mAP=\frac{1}{n}\sum_{k=n}^{k=1}AP_{k}$$

Finally, the number of correctly detected objects was measured for all trained models on each dataset, classifying detected objects in two categories: labelled objects that were detected (true positives or TP) and correctly detected objects that were not part of the annotations made for the validation datasets (false positives or FP). This last category comprises the nanoparticles that were not labelled because they were either cropped at the edge of a given image or partially occluded by other NPs.

## 3. Results

### 3.1. Testing on the Source Images

Generating semi-arbitrary micrographs was designed to increase the data available for the training of the YOLO neural network, and as such, the first attempt to evaluate this approach was to test the trained detectors on the *source* set of images, these being the group from which some micrographs were selected to extract the image templates of nanocubes. The actual source images are a subset of the validation Dataset I, whilst Dataset II includes some SEM micrographs of the same Ag sample but depicting scenes with a greater nanoparticle count and little to no space left between them. The charts displayed on Figure 8 summarize the achieved results of running a detection on all the images of Dataset I, the maximum IoU value attained at the 10,000th training batch by the M3 model, 90.22%. Trained during a total of 100,040 epochs, the Jaccard coefficient varied greatly throughout the process, reaching less variation between iterations after the 170,000th training batch, as shown on Figure 8a. These results differ from those of Dataset II, where there is little variation between iterations on models M1, M2 and M3, with the second achieving the highest value of 88.32% at the 130,000th training batch. Model M4, on the other hand, displayed the lowest results throughout the training process as well as the highest variation between iterations (see Figure 9a).

For median average precision on Dataset I, the results show less variation between iterations but greater among models, Figure 8b shows models M1 and M3, both using YOLOv3-tiny architecture, behaving almost inversely to each other after the first quarter of the training process, with each obtaining mAP values of 56.46% and 53.28%, respectively. In contrast, models M2 and M4 presented similar behaviour after the 50,000th training batch. However, they differ greatly in their capacity to correctly classify detected objects with Model M2 reaching a mAP of 64.41%, while M4 obtained a maximum of 46.41%.

#### Detected Particles

Table 5 shows the percentage of correctly detected nanoparticles achieved by the trained models on the first validation dataset, the highest value, 77.68%, belonging to Model M2 after 140,000 training batches. That is, from 896 annotated objects, Model M2 correctly detected 505, plus an additional 240 NPs that were not labelled. For the second dataset, the percentage of correctly detected particles ranged between 13 and just below 25%, the highest value attained by Model M4 after 120,000 training batches yielding 24.90% of correctly detected particles, equivalent to 430 out of 1727 labelled NPs plus 62 which were not originally annotated (see Table 6).

### 3.2. Testing on a New Dataset

The third validation dataset, composed of micrographs of HA and Ag nanoparticles, was used to evaluate the performance of the trained models on images not belonging to the original Ag sample nor the NFFA SEM image repository. The charts displayed in Figure 10 summarize the achieved IoU and mAP values for the third validation dataset, in which the M4 model attained values of 67.54% and 35.32%, respectively.

For this third dataset, the trained models yielded similar results to the ones obtained for the first two, a disparity between the Jaccard coefficient and mAP, regardless of the model tested and, in the case of this dataset, the distance between the nanoparticles depicted in a given image.

#### Detected Particles

After testing the trained models on the third validation dataset, Model M4 yielded the highest percentage of correctly detected particles at 62.50% after 10,000 training batches, analogous to the detection of 40 out of 64 annotated objects plus 31 unlabelled nanoparticles, as shown in Table 7. This table also shows that for Models M1 and M2, the highest percentage was achieved at least twice after training with increasing numbers of training batches.

As shown in the graphs in Figure 8, Figure 9 and Figure 10, there is noticeable difference between the models based on Version 3 of the YOLO neural network and the two based on Version 4 which can be attributed to the improvements between these versions. The values of mAP display the greatest variation, particularly for Dataset II and stand in contrast with the results obtained for the Jaccard coefficient, where the latter is above the 80% mark. The mAP for the four models remained under 70% (see Figure 9b). The disparity between the Jaccard coefficient, mAP and the percentage of detected NPs seen in all datasets suggest that while most of the trained models are capable of detecting the nanoparticles present in a given image, they display moderate success in being able to correctly categorize them. This is tied to the similarity between the training datasets and the validation ones.

In Figure 8, Figure 9 and Figure 10, it is observed that as the training batches increase, there is an instability in the detection of objects. This is explained considering that the complexity of the images used in the testing datasets increased as described below. It is important to mention that it was performed in this way since it is expected that any user who has micrographs obtained in a SEM, regardless of the quality of the images, can use these YOLO-based models to support the characterization of their materials.

Dataset I contains images very similar to the ones used to extract the templates for synthetic image generation. Not only are distinguishing features such as the contrast between the edges of a nanocube it faces very similar, but their position and numbers across the image resemble the scattering these nanostructures display when viewed through a scanning electron microscope, hence the high levels of detection on this dataset. Next, we have the images from the second validation dataset, which display a higher number of Ag nanoparticles per image, as well as these being much closer to each other and even having some nanocubes on top of others. This coupled with the refinement of the images during capture, which removed the bright edges on most of the observed nanocubes, and YOLO’s difficulty to detect objects that are within close proximity to each other, makes the whole set of annotated nanocubes different enough from the ones used during training to make the trained models miss most of the particles in the dataset.

Finally, the characteristics of the electron microscopes, such as maximum magnification, resolution of both SE and BSE detectors and electron source, used to capture the images used for training are different from the one used for validation. The micrographs depicting Ag nanocubes from Datasets I and II, along with the ones belonging to the NFFA repository were captured at a greater magnification than the one allowed by the SEM equipment used for the images from Dataset III. These differences, along with the variables added by the sample preparation methods used prior to characterization by electron microscopy, make the particles easier or harder to identify to both the human eye and the trained neural network models. The relatively low resolution of the latter microscope produced noise around the nanoparticles which made it difficult, even for a human expert, to distinguish a nanocube from a quasi-spherical particle.

## 4. Discussion

In this paper, we presented a simple method to generate images that resemble SEM micrographs of scattered Ag nanoparticles, allowing the training of a neural network based on two versions of the YOLO tiny architecture. The resulting models were evaluated on three validation datasets and proved to be capable of detecting nanoparticles with cubical and quasi-spherical morphologies, even when these were agglomerated in dense groups or blurred by some noise. However, a visible disparity existed between the values for the Jaccard coefficient and the median average precision yielded by all the models, with the first being considerably higher than the latter. This behaviour is a consequence of the little variation between the artificially generated images as well as the source images used, which in turn limited the data with scenes of nanocubes close to each other and resulted in the models being able to detect nanoparticles but having a poor ability to classify them correctly. This was most notorious for the images of the validation Dataset III, which was characterized by noisy images, as a result of the limited resolution of the scanning electron microscope available to us for the capture of images of the Ag and HA samples.

To improve these results, it would be necessary to modify the program that generates the artificial images so their resemblance to real SEM micrographs that include features such as agglomerated particles, lower contrast between nanocubes and the background and moderately noisy images is more precise. Additionally, the trained models show a lower variation between training batches, both for Jaccard coefficient and mAP, after 160,000 batches (see Figure 8, Figure 9 and Figure 10) which suggests that the improvement of these detectors will require a larger dataset of images as well as a longer training process.

## Figures and Tables

**Figure 1 nanomaterials-12-01818-f001:**
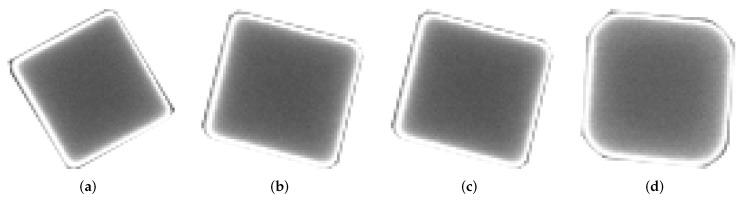
Manually segmented images of Ag nanocubes used as template to generate new images. These four templates, taken from the micrograph in Figure 2, constitute the second pattern used in this work: (**a**) Template #1 of an Ag nanocube; (**b**) Template #2 of an Ag nanocube; (**c**) Template #3 of an Ag nanocube; and (**d**) Template #4 of an Ag nanocube.

**Figure 2 nanomaterials-12-01818-f002:**
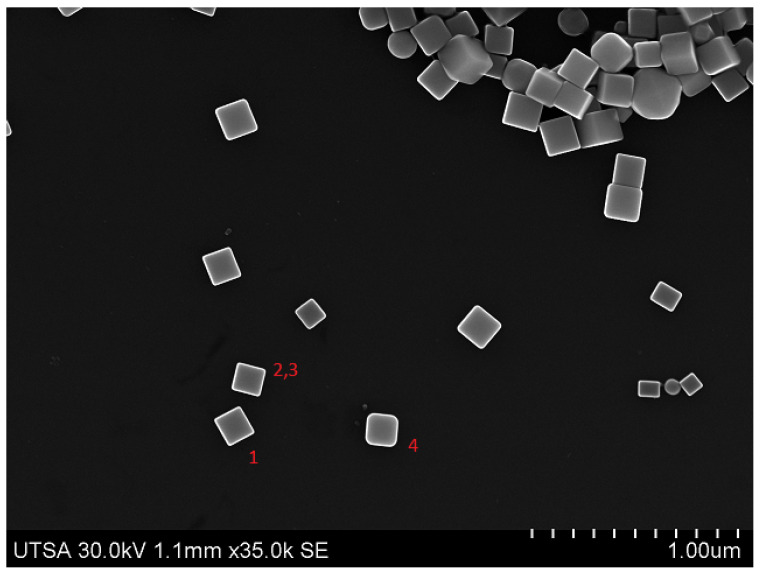
An example of an original SEM micrograph of Ag nanocubes from which templates were extracted.

**Figure 3 nanomaterials-12-01818-f003:**
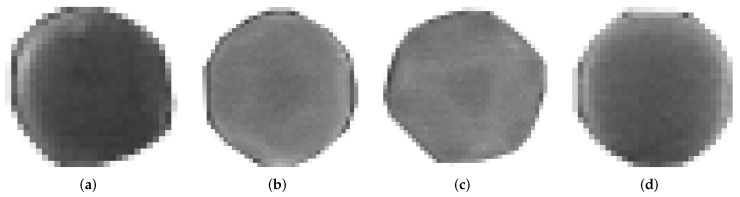
Manually segmented images of Ag quasi-spheres used as templates to generate new images. These four templates, taken from the micrograph in Figure 2 are part of a set of images used in the fourth pattern in this work: (**a**) Template #1 of an Ag quasi-sphere; (**b**) Template #2 of an Ag quasi-sphere; (**c**) Template #3 of an Ag quasi-sphere; (**d**) Template #4 of an Ag quasi-sphere.

**Figure 4 nanomaterials-12-01818-f004:**
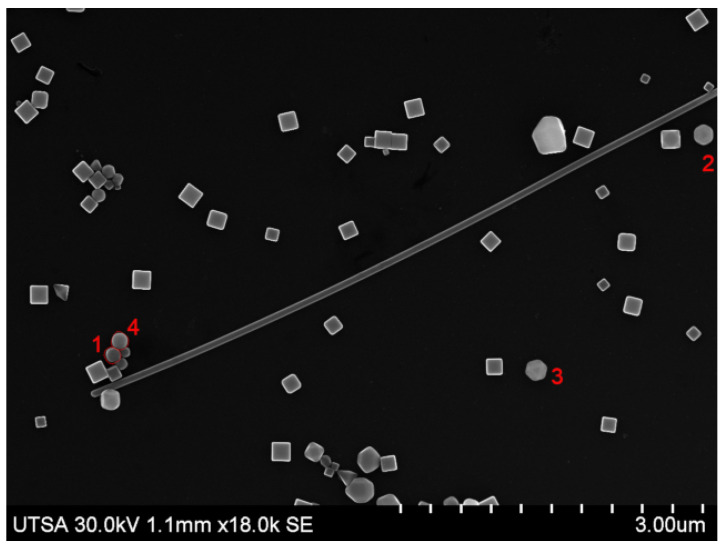
An example of an original SEM micrograph of Ag nanocubes and quasi-spheres, among other structures, from which templates of both morphologies were extracted.

**Figure 5 nanomaterials-12-01818-f005:**
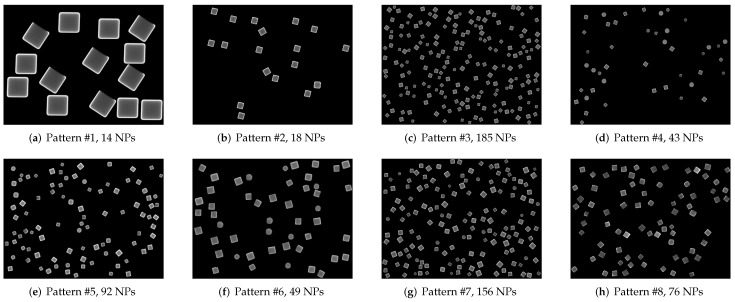
Synthetic images generated using the different patterns with varying numbers of scattered nanoparticles. Unlike the original micrographs, these images do not contain overlapping particles.

**Figure 6 nanomaterials-12-01818-f006:**
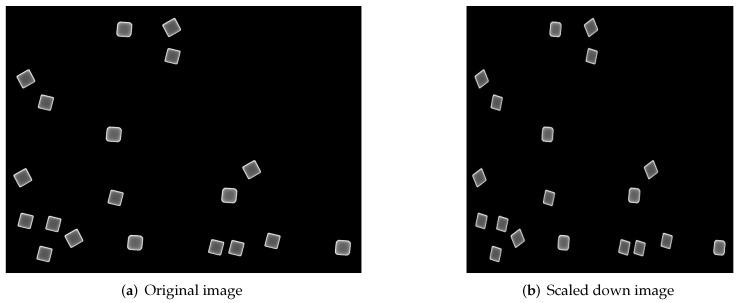
An example of a synthetic image used for training in its original and scaled down resolutions, the latter being the one the size of the input layer for the YOLO neural network.

**Figure 7 nanomaterials-12-01818-f007:**
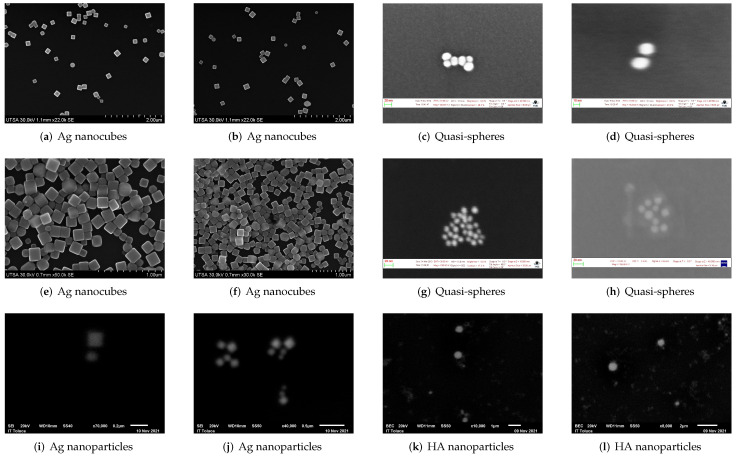
Some images from the three datasets used for the evaluation of the trained models. Dataset I (first row); Dataset II (second row); Dataset III (third row).

**Figure 8 nanomaterials-12-01818-f008:**
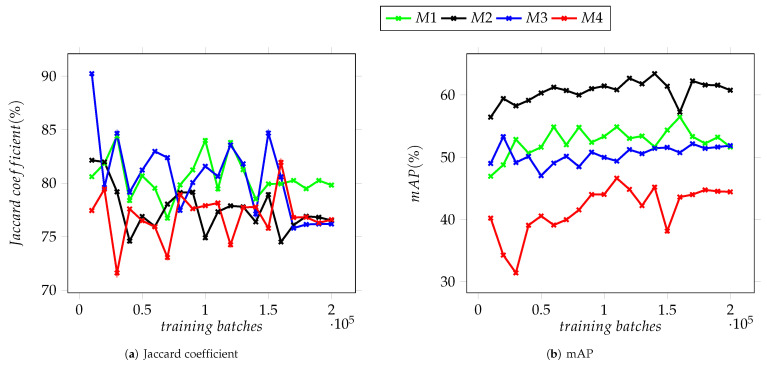
Average Jaccard coefficient and mAP for the trained detection models on validation Dataset I.

**Figure 9 nanomaterials-12-01818-f009:**
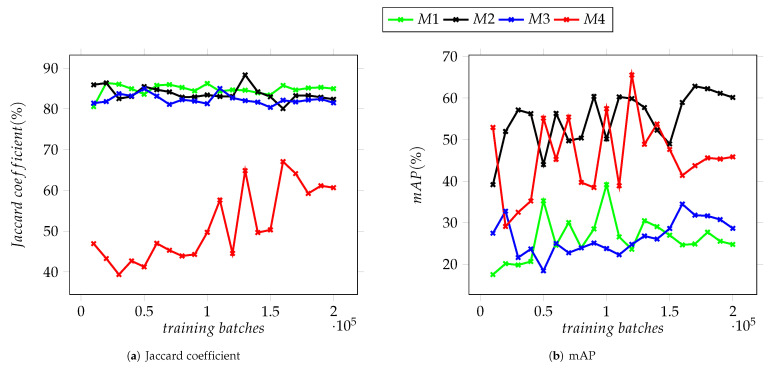
Average Jaccard coefficient and mAP for the trained detection models on validation Dataset II.

**Figure 10 nanomaterials-12-01818-f010:**
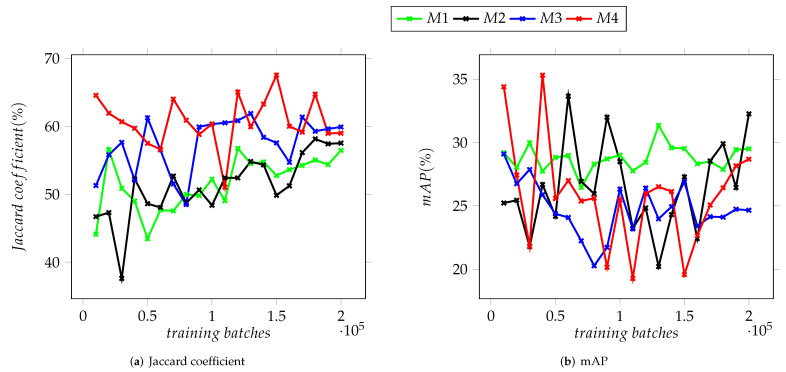
Average Jaccard coefficient and mAP for the trained detection models on validation Dataset III.

**Table 1 nanomaterials-12-01818-t001:** A list of patterns used to create artificial images resembling SEM micrographs.

Pattern Number	Number of Templates	Resolution (Pixels)	Morphologies	Avg. Particle Length
1	2	640 × 480	Cubes	0.146 μm
2	4	1280 × 960	Cubes	0.13 μm
3	6	1280 × 960	Cubes	0.13 μm
4	10	1280 × 960	Cubes & quasi-spheres	0.15 μm
5	31	640 × 480	Cubes & quasi-spheres	0.15 μm
6	17	640 × 480	Cubes & quasi-spheres	0.15 μm
7	29	640 × 480	Cubes & quasi-spheres	0.14 μm
8	57	1280 × 960	Cubes	0.15 μm

**Table 2 nanomaterials-12-01818-t002:** Distribution of object classes in the selected training datasets.

Set	Cube Class	Quasi-Sphere Class	Total Images
A	450	1000	105
B	1025	1025	112

**Table 3 nanomaterials-12-01818-t003:** A list of the four trained models, both tiny architectures of the YOLO neural network are trained on the two available datasets.

Model	Architecture	Trained on Dataset
M1	YOLOv3-tiny	A
M2	YOLOv4-tiny	A
M3	YOLOv3-tiny	B
M4	YOLOv4-tiny	B

**Table 4 nanomaterials-12-01818-t004:** Validation datasets used to evaluate the performance of the trained models.

Set	Labelled Cube Class Objects	Labelled Quasi-Sphere Class Objects	Total Labelled Objects	Total Images
I	553	343	896	20
II	1104	623	1727	20
III	17	47	64	20

**Table 5 nanomaterials-12-01818-t005:** Detection results for all models on the validation Dataset I.

Model	Correctly Detected NPs (%)	Training Batch
M1	67.19%	130,000
M2	77.68%	140,000
M3	62.72%	20,000
M4	53.24%	110,000

**Table 6 nanomaterials-12-01818-t006:** Detection results for all models on the validation Dataset II.

Model	Correctly Detected NPs (%)	Training Batch
M1	15.63%	100,000
M2	24.26%	170,000
M3	13.14%	160,000
M4	24.90%	120,000

**Table 7 nanomaterials-12-01818-t007:** Detection results for all models on the validation Dataset III.

Model	Correctly Detected NPs (%)	Training Batch
M1	54.69%	10,000–30,000, 60,000 & 90,000
M2	57.81%	180,000 & 200,000
M3	59.38%	10,000
M4	62.50%	10,000

## Data Availability

The source code for the nanocube generator program described in this document and the validation datasets used to evaluate the models are available at https://gitlab.com/jlcan/nanocube-generator (accessed on 21 April 2022) and https://disk.yandex.com/d/lDQVR2FPkqgW9w (accessed on 21 April 2022), respectively.

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
