# Peer review of "Nanoparticle Detection on SEM Images Using a Neural Network and Semi-Synthetic Training Data"

_nanomaterials, 2022, doi:10.3390/nano12111818_

Round 1

Reviewer 1 Report

The authors have adopted all previous suggestions in a good way. The manuscript is now in a version ready for publication.

Only one typo in line 168: should be "also" instead of "alse"

Author Response

Line 168: should be “also” instead of “alse”.

The typo was corrected

Reviewer 2 Report

The work "Nanoparticle Detection on SEM Images Using a Neural Network and Semi-Synthetic Training Data" studied automated particle detection in SEM images. The same works demonstrate significant interest both from the applied point of view and in the development of image processing methods using neural networks. Despite the current work requires significant improvements, I recommend accepting it for publication in the journal Nanomaterials after changes.

1) What is the advantage of YOLO if there are other working algorithms? Write it in detail in the introduction.
2) Why is the same particles used in two different templates (2 and 3)?
3) What templates were used to generate artificial images with spherical particles?
4) If the images were scaled up to 416*416 pixels for analysis, this certainly led to both the loss of information in object detection and the loss of information about the shape. It is necessary to prove that this procedure is no worse than image cropping or full-scale image processing.
5) From Figures 5-7, it doesn't look like the detection is stable. With an increase in training batches, the detection parameters strongly fluctuate. The authors need to explain this behaviour in more detail in the text.
6) To evaluate the effectiveness of the algorithm, it is necessary to provide detailed data on both the percentage of particles detected and the percentage of particles correctly recognized by shape.

Author Response

We have addressed the points raised by the reviewer in the attached document.

Round 2

Reviewer 2 Report

1) The answer to the question about the same templates should be added to the text.

2) Add in the text a description of method how the image was changed for processing by the neural network. Has the image been cropped or stretched? Have the real shapes of objects also changed, and how does this effect on the detection of particles?

3) A clear answer is needed why an increase in the number of training batches does not lead to an increase in the stability of detection of objects. Otherwise, how the obtained results can be applied to the other datasets, instead of selected training set?

Author Response

Addressing the questions and remarks made:

  1. The answer to the question about the same templates should be added to the text.

    The explanation was included in the text in lines 103 – 106.

  2. Add in the text a description of method how the image was changed for processing by the neural network. Has the image been cropped or stretched? Have the real shapes of objects also changed?, and how does this effect on the detection of particles?

    An explanation has been added to the manuscript in lines 169 – 176 and Figure 6.

  3. A clear answer is needed why an increase in the number of training batches does not lead to an increase in the stability of detection of objects. Otherwise, how the obtained results can be applied to the other datasets, instead of selected training set?

    An explanation was added at lines 267 – 272 & 311 – 317.